# A Facile Strategy toward the Preparation of a High-Performance Polyamide TFC Membrane with a CA/PVDF Support Layer

**DOI:** 10.3390/nano12244496

**Published:** 2022-12-19

**Authors:** Feng Liu, Yanyan Li, Lun Han, Zhenzhen Xu, Yuqi Zhou, Bingyao Deng, Jian Xing

**Affiliations:** 1School of Textile and Garment, Anhui Polytechnic University, Wuhu 241000, China; 2Advanced Fiber Materials Engineering Research Center of Anhui Province, Anhui Polytechnic University, Wuhu 241000, China; 3College of Materials and Textile Engineering, Jiaxing University, Jiaxing 314001, China; 4Laboratory for Advanced Nonwoven Technology, Key Laboratory of Eco-Textiles, Ministry of Education, Jiangnan University, Wuxi 214122, China

**Keywords:** cellulose acetate, poly(vinylidene fluoride), support layer, hydrophilic modification, interfacial polymerization, nanofiltration

## Abstract

In this study, polyamide (PA) thin-film composite (TFC) nanofiltration membranes were fabricated via interfacial polymerization on cellulose acetate (CA)/poly(vinylidene fluoride) (PVDF) support layers. Several types of CA/PVDF supports were prepared via the phase inversion method. With increasing CA, the PVDF membrane surface pore size decreased and hydrophilicity increased. The effect of the support properties on the performance and formation mechanism of PA films was systematically investigated via an interfacial polymerization (IP) process. The permselectivity of the resulting TFC membranes was evaluated using a MgSO_4_ solution. The results show that the desired polyamide TFC membrane exhibited excellent water flux (6.56 L/(m^2^·h·bar)) and bivalent salt ion rejection (>97%). One aim of this study is to explore how the support of CA/PVDF influences the IP process and the performance of PA film.

## 1. Introduction

In recent decades, science and technology have progressed and the development of polymer materials and polymeric separation membranes has become a new research hotspot [1,2,3,4]. Membrane processes have a wide range of applications in our lives, such as in paper, food processing, pharmaceutical production, and chemical and wastewater treatment, because they possess desired filtration properties [5,6,7]. Nowadays, the global imbalance in water resource distribution and the scarcity of fresh water sources has resulted in the urgent demand for new water-supply sources. In recent years, to solve this issue, people have started focusing their attention on seawater desalination and sewage reuse. The membrane process is an effective means to obtain fresh water from seawater and sewage, which is a practical application. Nanofiltration (NF) and reverse osmosis (RO) membranes have attracted worldwide interest in the field of desalination and wastewater recycling [8,9,10,11,12].

NF and RO are typical membrane processes which utilize hydraulic pressure for separation to obtain clean water. At present, the polyamide (PA) thin-film composite (TFC) membrane is the main type of NF and RO membrane. PA TFC membranes are composed of an ultrathin PA layer, a porous ultrafiltration middle layer, and an underlying layer of polyethylene terephthalate (PET) nonwoven fabric. The PA layer forms on the surface of the porous support layer via polycondensation of piperazine (PIP) and trimesoyl chloride (TMC). The thickness of the PA film ranges from a few tens to hundreds of nanometers, and the average pore size is less than 2 nm. PA film, as the active layer, plays a critical role in the permselectivity of TFC membranes. The ultrafiltration support layer of TFC membranes provides a place for interfacial polymerization, which affects the degree of cross-linking of the PA chain, as well as the thickness of the PA film [13,14,15,16]. At present, polysulfone (PSF) and polyethersulfone (PES) ultrafiltration membranes are the dominant candidates of the middle support layer. This is attributed to their relative hydrophobicity, high chemical resistance to benzene and hexane, strong heat-aging resistance, and good environmental endurance. The underlying layer of the PET nonwoven fabric provides mechanical support and can withstand strong hydraulic pressure in practical application processes.

PA is a dense thin film, and its inter-channels allow water molecules to pass through and salt ions to be excluded. The trade-off between the permeating flux and rejection is the major challenge of PA TFC membranes. Generally, the higher the water flux, the lower the salt rejection for TFC membranes, and vice versa. It is common knowledge that a slight change in PA film properties might significantly improve the overall permselectivity property of the TFC membrane. Numerous studies have explored achieving this goal in recent decades, but most research has focused on surface modification and optimization of the structure of PA film and including nanoparticle additives; surface chemical grafting; and the investigation of polymerization mechanisms. With the depth of research, it is realized that the support layer is also a vital factor in the IP process, and consequently, affects the performance of the PA layer.

Nowadays, the boom in the advanced chemical industry has caused serious water pollution. Wastewater contains numerous organic chemical reagents that result in PA TFC membranes with PES or PSF support layers suffering damage. The PES and PSF membranes are poorly resistant to some common chemicals, including aromatic hydrocarbons, ketones, ethers, and esters. To adapt the conditions of effluents and enlarge the application of TFC membranes, seeking a membrane material with better and more comprehensive properties than PES and PSF membranes has become a new research direction. In order to explore new support layer materials with superior performance, scientists have applied a variety of polymer materials for fabricating the porous support layer. The Hsiao group fabricated poly(acrylonitrile-co-acrylic acid)(PAN–AA)/polyacrylonitrile (PAN) nanofibrous supporting layers. The double-layer structure can capture and reserve abundant PIP monomers to facilitate interfacial polymerization, and TMC monomers. The resultant membrane showed a high water flux (64 Lm^−2^h^−1^) for Mg_2_SO_4_ solution. Gorgojo et al. used solvent-stable cross-linking polyimide as a support layer to fabricate PA TFC membranes, and the water flux was increased from 2 Lm^−2^h^−1^ to 16 Lm^−2^h^−1^. Pan et al. modified the hydrophilicity of a PP ultrafiltration membrane surface via UV-induced grafting of acrylic acid (AAc). However, the fabrication methods of these polymer support layers are complex and costly. A facile strategy for the preparation of polymer support layers is what researchers have been pursuing.

Polyvinylidene fluoride (PVDF) is an excellent membrane material for fabricated microfiltration and ultrafiltration membranes and has been used in large-scale commercial applications due to its excellent chemical resistance, good thermal and mechanical properties, and ease of manufacture [17,18,19,20]. However, the inherent hydrophobicity of PVDF limits its application as a support layer for PA TFC membranes. It is difficult to wet the PVDF membrane in a short time, which may result in the production of defective PA film. In order to make PVDF a support layer for TFC membrane use, it is necessary to improve the hydrophilicity of the PVDF membrane. According to related literature reports, many works have been conducted to improve the hydrophilicity of PVDF, including chemical grafting and plasma modification to introduce the hydrophilic groups to the PVDF membrane surface. A traditional PVDF casting membrane is prepared using the phase inversion method, and mixing this with the hydrophilic polymer material is a simple and effective way to improve the hydrophilicity of membranes.

Cellulose acetate (CA) is the acetate ester of cellulose and has a good application performance in membrane processes due to it is environmentally friendly nature, good biocompatibility, low price, and high hydrophilicity, as well as its high chlorine tolerance [21,22,23,24,25]. Although in some reports, CA is used as a support layer for PA TFC membranes, its mechanical strength and thermal stability need further improvement. In this work, we fabricated a CA/PVDF blend support layer using the one-step phase inversion method. To the best of our knowledge, there are no reports on using a CA/PVDF membrane as a support in the fabrication of PA TFC membranes. Our main aim in blending CA and PVDF is to overcome the development drawbacks of traditional PA TFC membranes. The results of this study indicate that the content of CA has a significant effect on the properties of the CA/PVDF membrane, such as its surface hydrophilicity, mechanical strength, pore size, and porosity, and further affects the PA film formation mechanism, surface morphology, and permselectivity of the PA TFC membrane.

## 2. Materials and Methods

### 2.1. Materials

Polyvinylidene fluoride (PVDF, Solef 6010, Mw ≈ 320,000) was purchased from Solef (Suzhou, China) and cellulose acetate (CA, Mw ≈ 100,000) was purchased from Sigma-Aldrich (Shanghai, China). N,N-Dimethylacetamide (DMAc, ReagentPlus, 99%) was provided by the China National Pharmaceutical Group Corporation (Beijing, China). Piperazine (PIP, 99%) and trimesoyl chloride (TMC, 98%) were supplied by Sigma-Aldrich. Triethylamine (analytical reagent, TEA), sodium dodecyl sulfate (SDS, Acros), and N-hexane (analytical reagent, 95%) were purchased from Sigma-Aldrich. Deionized water was produced using a Milli-Q system (Millipore, Burlington, MA, USA).

### 2.2. Membrane Fabrication

#### 2.2.1. CA/PVDF Support Membrane

CA/PVDF ultrafiltration membranes were fabricated using the phase-inversion method. The polymer concentration of the casting solution was 18 wt%. The content of CA in the polymer was fixed at 0, 10, 20, 30, 40, and 50 wt%, respectively. Homogeneous solutions were cast on PET nonwoven fabrics use a lab-scale casting device at about a 200 μm thickness, and subsequently, immersed in deionized water at a temperature of 25 °C. The support layers were rinsed with deionized water to remove the residual solution, and then, stored in DI water before further testing. The supports with different CA content were labelled as support-0 to support-5, and a detailed description is shown in Table 1.

#### 2.2.2. Polyamide TFC Film

PA thin films were fabricated via an interfacial polymerization reaction. The CA/PVDF supports with PET fabric were fixed on a glass plate, and then, they were immersed in PIP aqueous solution for 5 min and picked up. The excess solution on the support surface was removed using a rubber roller. After this, the supports were immersed in TMC organic solution for 2 min to react. The PA TFC membranes were rinsed using n-hexane and heat-treated in an oven (at 70 °C) for 10 min. Details of this are given in our previous work [26]. The resulting PA TFC membranes were labelled as PA-0 to PA-5 and stored in deionized water before further characterization.

### 2.3. Membrane Characterization

The rheological properties of the dope solutions were tested using a rheometer (Anton Paar, Graz, Austria), with the shear rate in the range of 0.01 s^−1^ to 100 s^−1^ under a temperature of 25 °C. The surface and cross-section morphology of the supports and polyamide films were observed via scanning electron microscopy (SEM, su1510, Hitachi, Tokyo, Japan). All samples were dried in a vacuum drying oven and sputtered with platinum (Pt) before being observed using SEM. The surface roughness and pore size of the supports were evaluated using non-contact atomic force microscopy (AFM, Bruker Dimension Icon, Houston, TX, USA) and the measured area was 2 × 2 μm. X-ray Photoelectron Spectroscopy (XPS, Axis supra, Manchester, UK) was used to investigated the surface chemical compositions of all samples. The surface wettability of the supports was charactered using a water contact angle measuring instrument. Five locations for each sample were measured to calculate the average value. The crystalline structure of the supports was analyzed using X-ray diffraction (XRD, D2 Phaser, Stuttgart, Germany) and the scanning range of all samples was from 5° to 60°. The crystalline degree of the polymer membranes was measured via Differential Scanning Calorimetry (DSC, Q200, Los Angeles, CA, USA). The temperature ranged from 30 °C to 200 °C and the heating and cooling rate were both 10 °C/min. The mechanic strength was tested using a universal strength instrument with a stretching rate of 5 mm/min. Each support was tested at least five times and the average value calculated.

The surface porosity of the supports was analyzed via FESEM images using ImageJ software (National Institute of Mental Health, Detroit, MI, USA). The water flux of the supports was calculated using lab-scale dead-end flow equipment with an effective area of 3.14 cm^2^. The sample was fixed in the cell keep for 15 min at a transport pressure of 0.4 MPa until the water flux remained stable. The process was repeated at least five times to obtain an average value. The water flux was calculated using the follow equation [20,27]:(1)$$J=\frac{V}{A\times t}$$
where J is the water flux (L/m^2^·h), *V* is the permeation volume (L), and *A* and *t* are the effective filtration area (m^2^) and testing time (h), respectively.

A lab-scale cross-flow unit was used to measure the salt rejection and water permeability of the PA TFC membrane and the effective measured area was 3.14 cm^2^. The MgSO_2_ salt solution concentration of the feed was 2000 ppm and the actual applied pressure was 1 MPa. The salt ion rejection rate was calculated by measuring the conductivity of the feed and permeate solution. The rejection of the TFC membrane was calculated using the following equation:(2)$$R=\left(1-\frac{C_{p}}{C_{f}}\right)\times 100\%$$
where R (%) is the rejection, and *C_p_* and *C_f_* are the concentration of the feed and permeate solution, respectively.

## 3. Results and Discussion

### 3.1. Compatibility between PVDF and CA

#### 3.1.1. Solubility Compatibility Theory

PVDF and CA are two common polymer materials which are used commonly to fabricate porous membranes. They are both easily soluble in an organic solution of DMF. Solubility parameter theory is an effective method to judge the solubility of two polymers. Hansen indicates that the molecular interaction is determined by the hydrogen bond (*δ_h_*), dispersive interaction (*δ_d_*), and polar interaction (*δ_p_*). The solubility parameters can reflect the thermodynamic compatibility between polymers through the enthalpy of mixing; the solubility parameters of CA and PVDF are shown in Table 2. The formula for calculating it is as follows [28]:(3)$$\Delta H_{m}=V_{m}\left[{\left(\delta_{d,CA}-\delta_{d,PVDF}\right)}^{2}+{\left(\delta_{p,CA}-\delta_{p,PVDF}\right)}^{2}+{\left(\delta_{h,CA}-\delta_{h,PVDF}\right)}^{2}\right]v_{CA}v_{PVDF}$$
where *H**_m_* and *V_m_* are the enthalpy of mixing and the weight percent of the solvent of the blended solution, respectively. *v_CA_* and *v_PVDF_* are the weight percentages of the polymer. According to the formula, it is obvious that the weight fraction and solubility parameter play the key roles in the enthalpy of mixing. For the polar mixtures, the lower mixing enthalpy value indicates that the compatibility of polymers is higher. The *H**_m_* values of the different blend ratios of CA and PVDF are shown showed in Table 3. According to the calculated result, we find that the enthalpy of mixing increased as the blend ratio increased. The increasing trend is dramatic when the blend ratios are from 10:90 to 30:70. However, when the blend ratios are from 30:70 to 50:50, the changing trend is relatively moderate. Generally, the compatibility of CA and PVDF is good when *H**_m_* is low; moreover, increasing the CA amount can extend the segmental gap between the CA and PVDF chain, resulting in an increase in pore size and amorphous regions in the NIPS process. However, when *H**_m_* is higher, the compatibility of CA and PVDF become poorer, which might cause significant demixing in the blending process.

#### 3.1.2. Melting Point Method

Determination of the melting points of the blend system is another method used to judge the compatibility of polymers. Generally, a single polymer only has one melting point, and two polymers may be incompatible at the molecular level. In this case, it could be observed that the system had two different melting points in DSC curve. When the blend system only presents one melting point, it indicates that the two polymers possess good compatibility. It is observed that the CA/PVDF blend system only had one melting point (Appendix A). Therefore, we can conclude that CA and PVDF are miscible at the molecular level.

### 3.2. Characteristic Analysis of Supports

The morphology of the membranes cast on PET fabric was observed via scanning electron microscopy. The surface morphology of membranes with different ratios of CA is shown in Figure 1. From the images, we find that the pure PVDF membrane surface possess a larger pore size. The blended membrane’s surface pore size and porosity are effectively restrained with increasing CA content. ImageJ software was used to analyze the relevant parameters of the blended membranes, and the surface pore size distribution is exhibited in Figure 2. We can observe that the pure PVDF membrane has the largest pore size and pore size distribution. With increasing CA ratio of the blends, the pore size of the membrane surface decreases progressively. However, the surface porosity significantly increases when a small amount of CA is added into the polymer, and then, it starts to decrease. This phenomenon is attributes to the different hydrophilic and interactions between CA and PVDF. CA is a hydrophilic polymer and PVDF is a hydrophobic polymer [22,29]. Additionally, when the CA content increases, the surface roughness of the blended membranes presents an obvious change.

SEM images of various CA/PVDF cross-sections are shown in Figure 1. All CA/PVDF supports present an asymmetric structure, including a dense sponge-like top layer and a finger-like sub-layer. For support-2, support-3, and support-4, we can clearly observe the macrovoids and obvious finger-like structure. These phenomena might be attributed to the low miscibility between CA and water. Furthermore, it also can be found that increasing the CA content promotes the formation of the sponge-like structure on the top layer of the support. As the CA content increases, the finger-like structure gradually disappears on the top, and the sponge-like structure appears and gradually increases. This change is because of the viscosity change in the polymer solution, which increases the viscosity of the CA/PVDF solution and delays the exchange of the solvent and non-solvent [29,30]. To further confirm the impact of solution viscosity, the rheological testing results of the dope solution are shown in Figure 3a,b. When the ratios of CA and PVDF are more than 30:70, the dense top layer starts demixing. This corresponds well with the solubility parameters of CA and PVDF; the solubility of CA and PVDF worsens, and more CA enriches the surface of the membrane with a gradual increase in CA content.

AFM images of the CA/PVDF membrane surface are shown in Figure 4. These images accurately represent the morphology and structure of the surface. We can observe bright and dark areas in the two-dimensional images, with the dark areas indicating membrane surface pores and valleys and the bright areas indicating higher points. The pore size was measured on a 2 μm × 2 μm area of membrane using SPM software 1.5. When the CA ratio reaches 10% in the blends, the surface porosity increases significantly. This is attributed to the addition of CA improving the hydrophilicity of the blend system [31]. The hydrophilicity of the dope solution increases and promotes the diffusion exchange of the solvent and non-solvent, which leads to larger pores and higher porosity on the surface. However, when the ratio of CA is more than 10%, the viscosity of the dope solution gradually decreases, and the pore size and porosity of the membrane gradually decrease. This phenomenon can be explained by the solubility parameters in Section 3.1. Support-1 has the lowest mixing enthalpy, indicating that CA and PVDF are more compatible. The mixing enthalpy value is higher, and the compatibility of the polymer is poorer. The hydrophilic nature of CA causes the CA segments to migrate to the surface, resulting in a smaller pore size. The surface roughness of the various blended membranes presented obviously change with increasing CA concentration in the casting solution, as shown in Table 4. We can find that the average square root of the Rq values of the blended membranes are significantly different compared with those of the pure PVDF membrane. When the CA is added into the casting solution, the surface roughness instantly increases, and then, gradually decreases as the CA content increases. The significant change in roughness might be attributed to the surface porosity increase.

The chemical composition of the membrane surface was analyzed via XPS, and the results are shown in Figure 3d,e. PVDF and CA are both pure polymers; their membranes only contain a few pivotal peaks, and they are located at 280 to 289, 397, 530, and 685 eV, respectively. According to the curves of the image, it can be found that all the samples have the same peaks. The oxygen (O) and nitrogen (N) element originate from the pore-forming agent in the pure PVDF membrane. Upon analysis of the content of “O” and “F” on the membrane surface, it can be found that the content of “O” gradually increases and the content of “F” decreases as the CA content increases. this is attributed to the CA content increasing, as well as the increase in CA enriching the surface of the blended membranes. CA is a hydrophilic polymer which can effectively improve the hydrophilicity of the PVDF membranes. The dynamic water contact angle values of all the membranes are shown in Figure 3c. The results indicate that the surface wettability of the supports increase as the CA content increases. The wettability of support-1 is better compared with support-2 and support-3, which is attributed to support-1 having ultra-high porosity. Good wettability contributes to the diamine aqueous solution of the fully wetted membrane pores in the interfacial polymerization process.

The pure water flux of the supports was tested using the lab-scale dead-end flow device. All the samples were measured under the same conditions and with the hydraulic pressure of manipulation at 0.25 and 0.3 MPa; the results are shown in Figure 3f. It is found that support-1 and support-2 have higher water flux than support-0. This is attributed to the fact that a certain amount CA caused an increased in the hydrophilicity and porosity of the membrane surface. However, for the other supports, the water flux decreases when the CA content is more than 20 wt%, because the porosity of the supports deceases. This is in line with the result that we observed from the surface and cross-section images of the supports. Although the hydrophilicity of the surface increases, the pore size and porosity of the surface decrease and the thickness of the top dense layer increases, resulting in a water transport resistance increase.

The effect of the ratio of CA on the mechanical strength of the membranes was investigated through the stress–strain curves (Appendix A). According to the observation, we find that the CA content in the polymer has a significant effect on the elongation at the break in the blended membrane. Support-0 has the maximum elongation and the minimum breaking strength in the process of stretching. However, as the CA content increases, the elongation decreases from 16.2% to 2.1%. Additionally, the strength slightly increases. It is also indicated that the elasticity of the blended membrane decreases and the brittleness increases. This phenomenon might be attributed to the change in the aggregate structure of PVDF polymer. Relevant tests were conducted to confirm change in the structure of the PVDF molecular chain. As seen in the DSC test results (Appendix A), the addition of CA to the PVDF polymer breaks the large molecular chain’s regular structure in part of the PVDF, resulting in the crystallization zone decreasing. To further analyze the crystalline structure of the CA/PVDF blended membranes, XRD analysis was conducted (Appendix A). Compared to the crystallographic data of the various membranes, all membranes present a typical α structure, with three typical characteristic peaks at 2θ ≈ 18.6°, 20.0°, 27°, respectively. They are the (100), (110) and (022) crystallographic planes of PVDF, respectively.

### 3.3. Effects of Surface Properties of Supports on Polyamide Film Formation

PA film is formed on the surface of CA/PVDF supports, and the reaction scheme was depicted in our previous work [31]. The surface morphology of PA films is shown in Figure 5, is which we can clearly observe that all films present a typical “ridge and valley” structure. The PA nascent layer forms in a very short reaction time. As the reaction time increases, the solution constantly flows out from the support pores, causing nascent layer blending and wrinkling, which form a “ridge and valley” structure. However, there are differences in the “ridge and valley” structures of PA films on different supports. This is mainly attributed to the CA/PVDF surfaces’ physicochemical properties. In the case of support-1 with the high porosity and large pore size, as shown in Figure 1b1, the PA film consists of a ridge with a large size that has a dense array and is multi-layered. For supports with a small pore size and moderate porosity (support-2 and support-3), the growth tendency of the ridge structure is suppressed, and the PA film possesses a medium size and moderate array. Regarding supports with a small pores size and low porosity (support-4 and support-5), the PA film consists of a ridge with a small size and sparse array. A schematic diagram of the formation of the PA film on different supports is presented in Figure 6 [15]. Images of a TFC membrane structure cross-section are shown Figure 7, from which it is clear that the PA TFC membrane presents a three-layered structure. The thickness of the PA film presents a slight change. This might be explained by the fact that the porosity and pore size of the CA/PVDF surface decrease.

### 3.4. Separation Performance of TFC Membranes

The separation performance of TFC membranes was investigated using a MgSO_4_ solution, and the results are shown in Figure 8a. For TFC membranes with support-0 and support-1, both present high water flux and low salt rejection. The hydrophobic surface of support-0 is averse to aqueous solution with sufficient wetting membrane pores. Meanwhile, support-1, with high porosity, easily produces a large pore size, which might result in the PA layer collapsing in the operation process. According to the test results, we find that TFC-2 possesses the highest water flux, and TFC-3 the best salt rejection, compared with TFC-4 and TFC-5. This can be attributed to support-2 and support-3 having moderate hydrophilicity and optimal porosity. For the TFC-3, TFC-4, and TFC-5 membranes, the salt rejection ratio does not significantly change and remains above 97%, but the permeate flux gradually deceases. This can be explained by the degree of cross-linking in the PA increasing when the surface pore size decreases. Additionally, the water flux of the TFC membrane decreases as the ratio of CA in the support increases. This is also ascribed to the increase in the sponge-like structure’s thickness on the support membrane, resulting in water transport resistance significantly increasing.

Figure 8b shows the separation performance of the TFC membrane with support-3, prepared by varying the polymerization time while keeping the other parameters constant. The permeate flux significantly decreases and the rejection rate increases with increasing polymerization time from 1 min to 3 min. This tendency is attributed to the long reaction time, leading to the formation of a PA film with high cross-linking density and thickness. Furthermore, it was found that 2 min is the optimal reaction time for PA film formation. Additionally, to further understand the impact of TMC concentration on TFC membrane flux behavior, we fabricated three types of PA film on support-3, and the TMC concentrations were fixed at 0.2, 0.3, and 0.4 wt%, respectively. The results are shown in Figure 8c; the operation time was 6h, and it is fould that the permeate flux decreases with increasing TMC concentration. This result is consistent with the results previously reported in the literature. Increasing the concentration of TMC results in higher cross-linking density in the formed PA film; thus, the flux decreases [32,33]. Based on the above analysis, we can conclude that the optimal PA film was fabricated on support-3, with a TMC concentration of 0.2 wt% and a polymerization time of 2 min. This result was compared with other works from the literature, as shown in Table 5.

## 4. Conclusions

CA/PVDF blended membranes were fabricated via a non-solvent-induced phase and as middle supports for PA TFC membranes. The ratio of CA had a significant effect on the morphology and relevant properties of the blended membrane, including the hydrophilicity, surface porosity, pore size, and roughness. The hydrophilic nature of CA improved the wettability of the PVDF membrane. The surface pore size and porosity increased, and then, decrease as the CA content increased. Additionally, the CA ratio of the polymer system had a great influence on the solubility parameters between CA and PVDF, as well as the inner structure and mechanical strength of the supports.

Novel PA TFC membranes were synthesized via interfacial polymerization on CA/PVDF supports. The characteristic of support significantly influenced the separation performance of TFC membrane. A support with appropriate pore size, porosity, and hydrophilicity was imparted to the TFC membrane, with high retention performance and promising permeation flux. This study provides a new protocol for the design of high-performance TFC membranes for seawater desalination and wastewater treatment.

## Figures and Tables

**Figure 1 nanomaterials-12-04496-f001:**
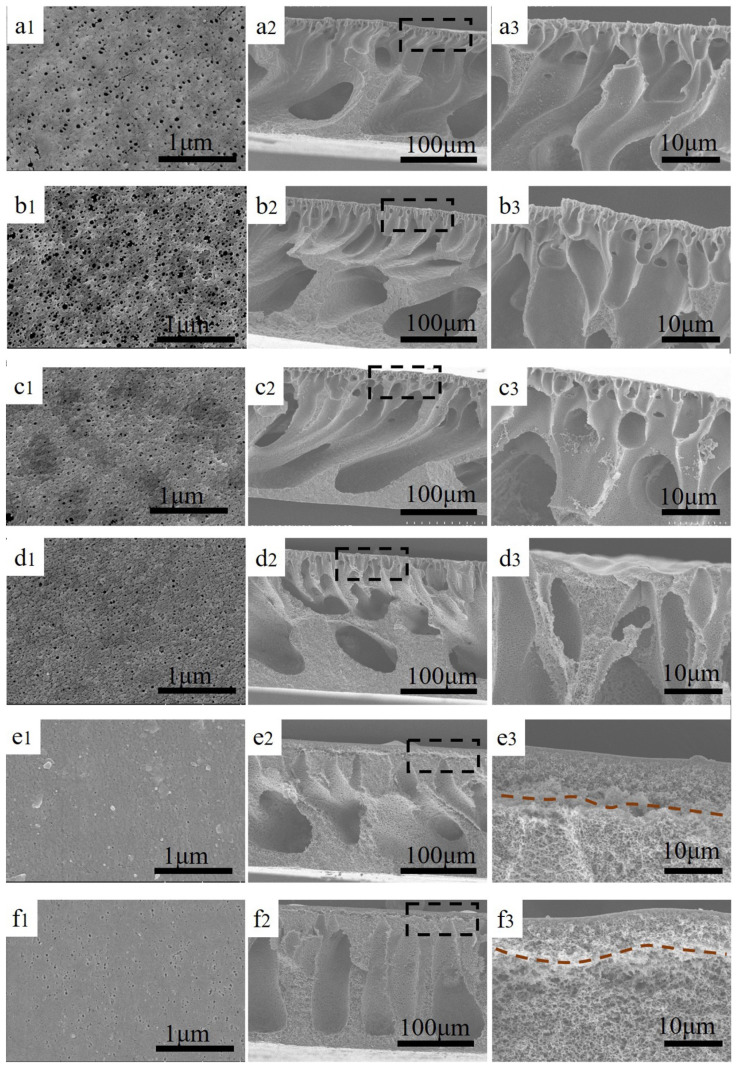
The surface and cross-sectional images of different supports: (**a1**–**a3**) support-0, (**b1**–**b3**) support-1, (**c1**–**c3**) support-2, (**d1**–**d3**) support-3, (**e1**–**e3**) support-4, (**f1**–**f3**) support-5.

**Figure 2 nanomaterials-12-04496-f002:**
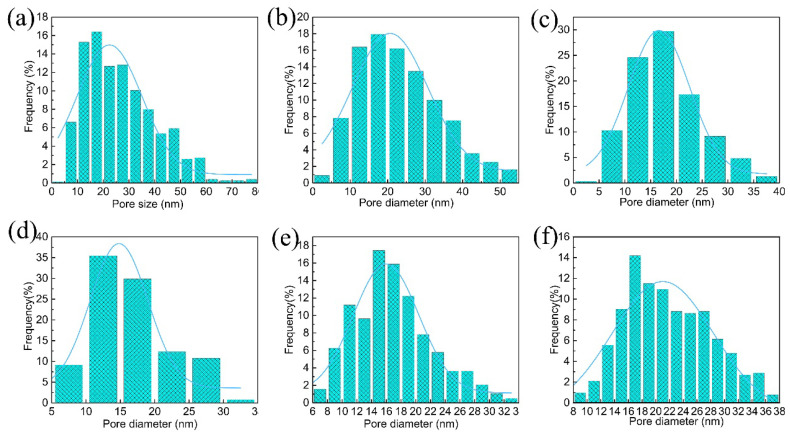
Pore size distribution of different supports: (**a**) support-0, (**b**) support-1, (**c**) support-2, (**d**) support-3, (**e**) support-4, (**f**) support-5.

**Figure 3 nanomaterials-12-04496-f003:**
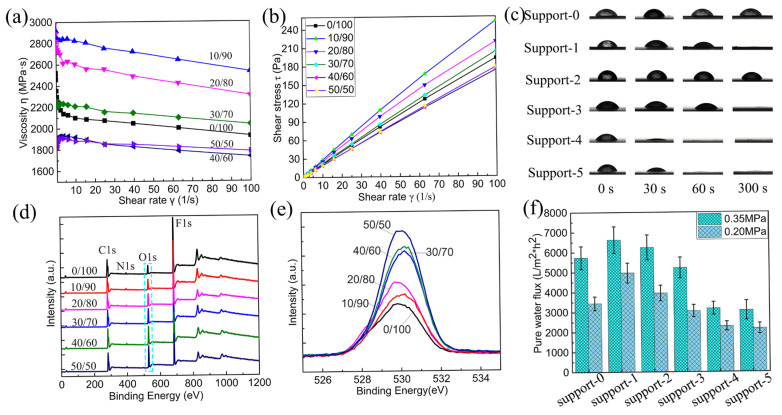
(**a**,**b**) The rheological properties of different dope solutions; (**c**) dynamic water contact angle of different supports; (**d**) XPS spectra of the different supports surfaces; (**e**) the O element spectra of different supports surfaces; (**f**) pure water flux of different supports.

**Figure 4 nanomaterials-12-04496-f004:**
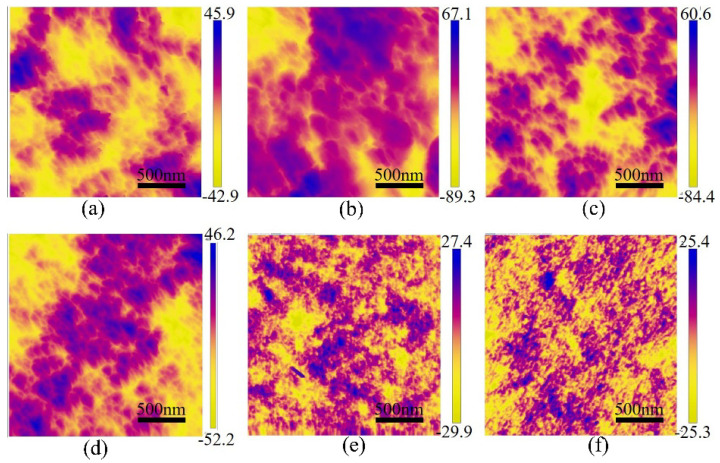
AFM images for the supports: (**a**) support-0, (**b**) support-1, (**c**) support-2, (**d**) support-3, (**e**) support-4, (**f**) support-5.

**Figure 5 nanomaterials-12-04496-f005:**
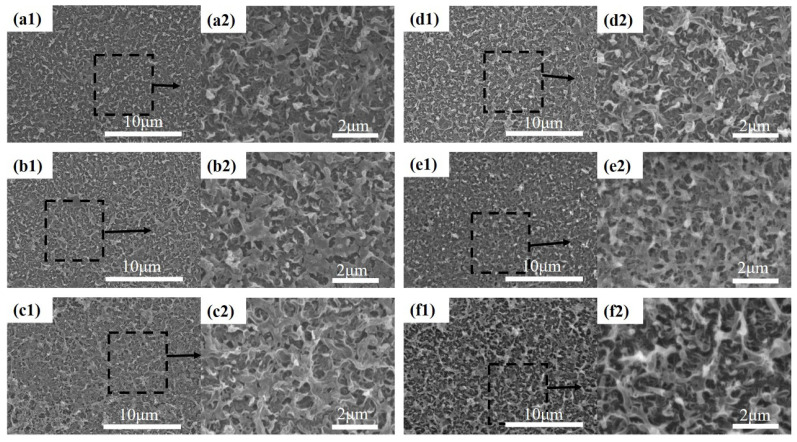
Surface SEM images of the polyamide TFC membranes with different supports: (**a1**,**a2**) TFC-0, (**b1**,**b2**) TFC-1, (**c1**,**c2**) TFC-2, (**d1**,**d2**) TFC-3, (**e1**,**e2**) TFC-4, (**f1**,**f2**) TFC-5.

**Figure 6 nanomaterials-12-04496-f006:**
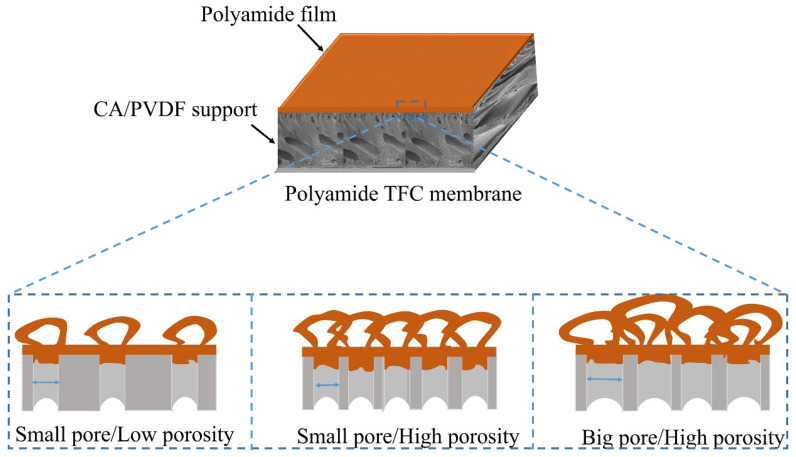
Schematic diagram of PA film formation on different supports.

**Figure 7 nanomaterials-12-04496-f007:**
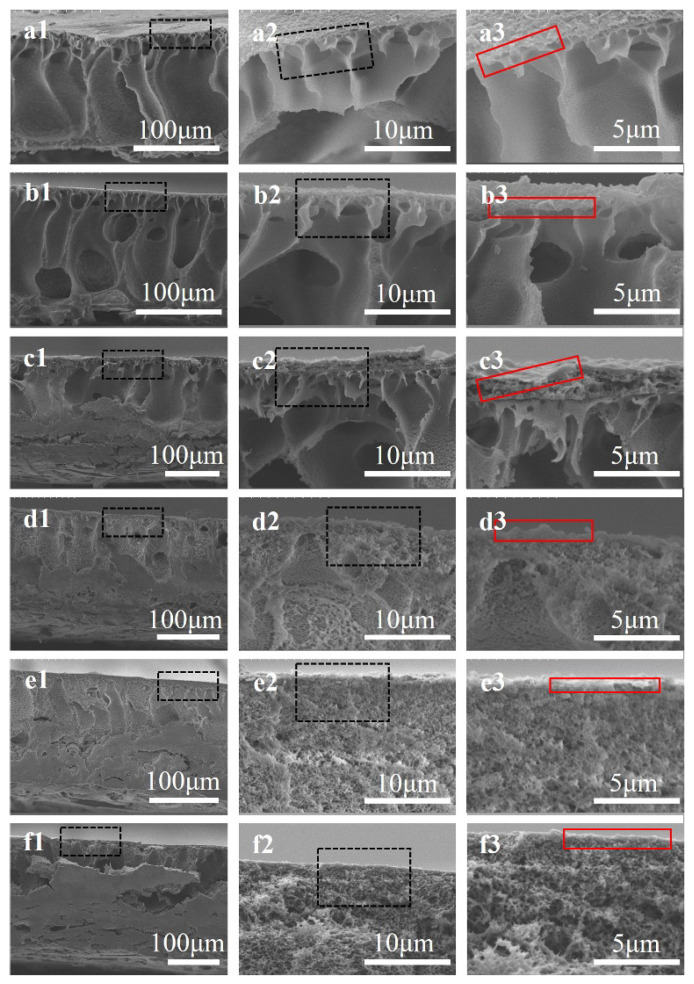
Cross-sectional SEM images of the polyamide TFC membranes with different supports: (**a1**–**a3**) TFC-0, (**b1**–**b3**) TFC-1, (**c1**–**c3**) TFC-2, (**d1**–**d3**) TFC-3, (**e1**–**e3**) TFC-4, (**f1**–**f3**) TFC-5.

**Figure 8 nanomaterials-12-04496-f008:**
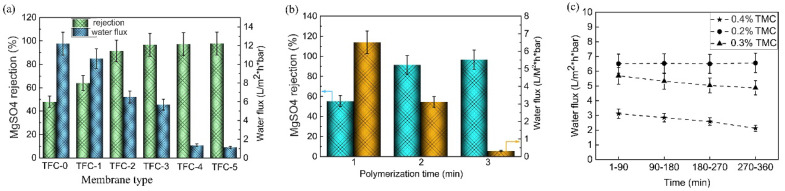
(**a**) Separation performance of the TFC membranes with different supports: (**b**) the effect of polymerization time on separation performance of the TFC membranes with support–3, MgSO_4_ rejection (blue), and water flux (brown); (**c**) the effect of TMC concentration on permeate flux of the TFC membranes with support-3.

**Table 1 nanomaterials-12-04496-t001:** The composition of the dope solution.

	Casting Solution Composition			
	CA (g)	PVDF (g)	PVP (g)	DMAc (mL)
Support-0	0	18	0.5	100
Support-1	1.8	16.2	0.5	100
Support-2	3.6	14.4	0.5	100
Support-3	5.4	12.6	0.5	100
Support-4	7.2	10.8	0.5	100
Support-5	9	9	0.5	100

**Table 2 nanomaterials-12-04496-t002:** Solubility parameters of CA and PVDF.

Material	*δ_d_*	*δ_p_*	*δ_h_*
CA	7.9	3.5	6.3
PVDF	17.2	12.5	9.2

**Table 3 nanomaterials-12-04496-t003:** The mixing enthalpy of prepared membranes.

CA/PVDF Composition	0/100	10/90	20/80	30/70	40/60	50/50
Δ*H_m_*	-	13.4	23.82	31.3	35.8	37.2

**Table 4 nanomaterials-12-04496-t004:** Surface roughness parameters and surface porosity of CA/PVDF membranes obtained from AFM and SEM images.

Membranes	Roughness Parameters			Surface Porosity (%)
	S_a_ (nm)	S_q_ (nm)	S*_z_* (nm)	
Support-0	10.1	12.7	83.2	7.27
Support-1	16.9	22.1	136	17.4
Support-2	13.8	17.5	118	9.2
Support-3	12.8	15.5	98.3	8.7
Support-4	6.34	8.04	73.7	4.8
Support-5	5.16	6.48	50.3	5.1

**Table 5 nanomaterials-12-04496-t005:** Comparison performance of fabricated TFC membrane with literature data.

Type of Polymer	Water Flux (Lm^−2^h^−1^)	Rejection (%)—Salt Ions	Reference
Copoly(phthalazinone biphenyl ether sulfone) (PPBES)	77	97.5—Na_2_SO_4_	[34]
Polyamide (PI)	16	93.8—NaCl	[35]
poly (phthalazione ether nitrile ketone) (PPENK)	57.9	98.4—Na_2_SO_4_	[36]
Polytetrafluoroethylene (PTFE)	420	93—Rhodamine B	[37]
Polyvinyl chloride (PVC)	11.3	58.1—NaCl	[38]
TiO_2_/PAN	63	99.4—Na_2_SO_4_	[39]
This work	65.6	97—MgSO_4_	

## Data Availability

Not Applicable.

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
