# Peer review of "A Facile Strategy toward the Preparation of a High-Performance Polyamide TFC Membrane with a CA/PVDF Support Layer"

_nanomaterials, 2022, doi:10.3390/nano12244496_

Round 1

Reviewer 1 Report

This work prepared TFC membranes on CA/PVDF supports. The support membranes were fabricated from different ratios of CA and PVDF. With the increase in CA, the support membrane has a larger hydrophilicity and smaller surface pore size. This change in surface property affects the TFC layer. The optimal TFC membrane obtained shows a reasonable flux and ion rejection. It is recommended for publication after some revisions.

1.     In 2.2.2, the detail reaction conditions should be provided. In addition, reference 25 uses MPD, but this work uses PIP.

2.     MgSO4 rejection was tested. What is salt rejection to monovalent ions, e.g. NaCl?

3.     It is claimed that “In the case of support-1 with high porosity and large pore size as shown in Fig. 1(b1), the PA film consists of ridge with big size”. The change in ridge size from big to middle and to small is not clear in the picture.

4.     The interfacial polymerization is fast. When 1, 2 and 3 mins of reaction time is compared, the water flux decreases a lot. Why is this so?

5.     English writing should be improved as there are quite some grammar mistakes. For example:

Page 9: “we can found that the CA content in polymer have significant effect”

Page 11: “we can found that TFC-2 possesses highest water flux”

Reviewer 2 Report

The authors have demonstrated the feasibility of how the support of CA/PVDF influence the IP process and the performance of PA film. Overall, this is a decent research paper and can attract the readers studying TFC membrane and NF process. Herewith I have attached my few comments which should be addressed:

1. Starting with the abstract, kindly highlight the novelty statement for better understanding and readability 

2. Coming to the next point, the introduction sectione seems to be articulated one. However, state of art of discussion is missing from the section. Kindly include a comparative study of previous research outcomes (materails/polymers/Flux/rejection) with the current one to show the vaibility of recent research. 

3. The lab setup of NF process must be shown in Materials and Methods section. Why magnesium sulphate has been used as feed stream solution?

4. The trend observed in Figure 3 isn't well supported by valid references. Please provide appropriate citations. 

5. While dealing with the membrane subject, the following information can be added for better readaility

(a) Thickness of the membrane since it may influence the flux in NF process.

(b) FTIR for the identification of organic chemical functional groups.

Reviewer 3 Report

1. The papers starts with an objective of overcoming the drawbacks of traditional PA TFC membrane . However as we go along the paper, 80 % of paper is dedicated to studying the properties of blend and only 10-15 % discusses the separation performance. Also since the paper is aimed at overcoming the drawback, it is suggested that the authors also compare the separation performance of membrane blends already reported in literature.

2. Secondly, due to poor construction of sentences and grammatical errors, it was extremely difficult to follow up with the arguments and results and discussions.

3. To the large extent, the paper reads like a report. There is no scientific driving motivation that has the rationale and the motivation. The chemistry should be discussed in details (why CA/PVDF blend was chosen, why is it expected to improve performance etc.). Mere reporting of some formulation which was not reported before does not constitute a research paper.

4. description of some characterizations methods and data discussion needs re-considerations. for example, for XPS, detailed description is missing, which level peaks are represented in Fig. 4e? (2s or 2p?). Why some shifts are observed in this peak? do you expect some contribution of O and C peaks from adventitious adventicious CO2? 

5. How long the membrane blend lasts for separation performance without failing? stability study should be present.

Reviewer 4 Report

The manuscript entitled " A facile strategy toward preparing high-performance polyamide TFC membrane with CA/PVDF support layer" studies different membrane properties for membranes modified by altering the support layer. The experiments are well analyzed and results are presented in an interesting way. In my opinion, the manuscript can be accepted as it is in the Nanomaterials. Few concerns from my side

1.      Error on page number 8 like dead--end flow this may be dead-end flow

2.      In figure 8, (a) y-axis rejection of what salt? please mention MgSO4 rejection similarly to figure 8 (b).

3.      What about the surface contact angles of the TFC membrane?

4.      What is the role of the CA/PVDF support layer for the TFC layer? Provide results published online and a comparison table to distinguish the results.

5.      The membrane performance obtained in this work must be compared with those of the membranes including nanomaterials reported previously, and the advantage of the membrane in this work must be clarified.

6.      Here are a few references for similar studies of PAN/PVDF as a substrate for UF and TFC membranes author may cite this to strengthen the manuscript. 

· J. Appl. Polym. Sci. 138.16 (2021): 50228. (https://doi.org/10.1002/app.50228)

· J. Appl. Polym. Sci.138.1 (2021): 49606. (https://doi.org/10.1002/app.49606)

·         Int. J. Adv. Res. Eng. 2017, 4, 2394–2444.

Round 2

Reviewer 2 Report

The authors have addressed all the queries with high scientific discussions. Therefore, the revised manuscript can be accepted in the present format. 

Reviewer 3 Report

Although the issue with English still exists in the manuscript, most f the comments were addressed.